# A Review of Disparities and Unmet Newborn Screening Needs over 33 Years in a Cohort of Mexican Patients with Inborn Errors of Intermediary Metabolism

**DOI:** 10.3390/ijns9040059

**Published:** 2023-10-19

**Authors:** Isabel Ibarra-González, Cynthia Fernández-Lainez, Marcela Vela-Amieva, Sara Guillén-López, Leticia Belmont-Martínez, Lizbeth López-Mejía, Rosa Itzel Carrillo-Nieto, Nidia Alejandra Guillén-Zaragoza

**Affiliations:** 1Laboratorio de Errores Innatos del Metabolismo y Tamiz, Instituto Nacional de Pediatría, Secretaría de Salud, Ciudad de México 04530, Mexico; 2Unidad de Genética de la Nutrición, Instituto de Investigaciones Biomédicas, Universidad Nacional Autónoma de México, Ciudad de México 04510, Mexico; 3Facultad de Química, Universidad Nacional Autónoma de México, Ciudad de México 04510, Mexico

**Keywords:** neonatal screening, monogenic disorders, clinical diagnosis, rare diseases, infant mortality, intellectual disability

## Abstract

Advances in an early diagnosis by expanded newborn screening (NBS) have been achieved mainly in developed countries, while populations of middle- and low-income countries have poor access, leading to disparities. Expanded NBS in Mexico is not mandatory. Herein, we present an overview of the differences and unmet NBS needs of a group of Mexican patients with inborn errors of intermediary metabolism (IEiM), emphasizing the odyssey experienced to reach a diagnosis. We conducted a retrospective observational study of a historical cohort of patients with IEiM from a national reference center. A total of 924 patients with IEiM were included. Although 72.5% of the diseases identified are detectable by expanded NBS, only 35.4% of the patients were screened. The mortality in the unscreened group was almost two-fold higher than that in the screened group. Patients experienced a median diagnostic delay of 4 months, which is unacceptably long considering that to prevent disability and death, these disorders must be treated in the first days of life. Patients had to travel long distances to our reference center, contributing to their unacceptable diagnostic odyssey. This study highlights the urgent need to have an updated, expanded NBS program with adequate follow up in Mexico and promote the creation of regional medical care centers. We also provide compelling evidence that could prove valuable to decision makers overseeing public health initiatives for individuals impacted by IEiM from middle- and low-income countries.

## 1. Introduction

Inborn errors of metabolism (IEM) constitute a significant group of rare genetic diseases affecting individuals of all ethnicities and ages [1]. IEM genetic variants are rare in the population, and children with these disorders exhibit a high morbidity risk, including intellectual and motor disability and premature mortality [2,3,4].

Inborn errors of intermediary metabolism (IEiM) are a subgroup of IEM comprising numerous conditions related to catabolism, anabolism, energy use, storage, and generation. IEiM affects thousands of proteins, mainly enzymes or transporters [5,6]. Many of these diseases are treatable with different approaches, such as dietary, pharmacological, enzyme replacement, and surgical (liver and liver–kidney transplant) therapies; gene replacement therapies will soon be available [7,8].

A significant number of IEiMs are associated with well-known biomarkers (amino acids, acylcarnitines, succinylacetone, or organic acids) that can be quantified in biological fluids, i.e., blood and urine, using robust methodologies, such as tandem mass spectrometry (MS/MS), high-performance liquid chromatography (HPLC), or gas-chromatography–mass-spectrometry (GC-MS) [9,10]. The quantification of these biomarkers has allowed the diagnosis of symptomatic patients (high-risk screening) and forms the basis for presymptomatic expanded newborn screening (NBS) programs [11]. Unfortunately, NBS programs that include the detection of IEiM by MS/MS still need to be universal. Latin American patients affected by these disorders have been largely under-represented in worldwide studies due to, among other reasons, a lack of awareness of these disorders among health personnel, leading to an underdiagnosis. Additionally, Latin American NBS programs are limited to the detection of only a few diseases [12] and have limited access to metabolic and genetic tests, such as Sanger sequencing, whole exome sequencing (WES), and whole genome sequencing (WGS) [13,14]. In Mexico, the mandatory NBS panel of the Ministry of Health, which attends most of the births in Mexico, comprises only six diseases: congenital hypothyroidism, phenylketonuria (HPA/PKU), congenital adrenal hyperplasia, galactosemia (GALAC), cystic fibrosis, and glucose-6-phosphate dehydrogenase deficiency [15]. In addition, there is a large variability in the NBS panels between the institutions constituting the national health system [16].

Although rare diseases indiscriminately impact all socioeconomic, geographic, and ethnic groups [17], advances in genetics, including diagnoses, epidemiology, molecular spectrum, management, therapies, and efficient NBS programs, have been established predominantly in high-income countries with populations of European ancestry, mainly because the study of patients with rare disorders requires infrastructure, permanent financial funding, extensive expertise, and public health policies [18,19].

Few works have been published on the epidemiology of IEiM in Mexico, and most cover regional or local experiences regarding specific disorders [20,21,22,23,24,25,26,27,28]. Therefore, Mexican patients with IEiM are under-represented in the current literature. In this observational study at our national reference center with 33 years of experience, we overviewed the epidemiological characteristics of a historical cohort of Mexican patients, emphasizing the disease frequency, geographical distribution, gaps in diagnoses and treatment, unmet diagnostic needs, diagnostic odyssey, and difficulties in traveling to the center among the included patients.

## 2. Methods

### 2.1. Study Design

We conducted a retrospective observational study of a historical cohort of patients with IEiM.

### 2.2. Setting Location

A clinic devoted to the medical care of patients with IEiM was established in 1990 at the National Institute of Pediatrics (INP, by its abbreviation in Spanish) in Mexico City, a high-specialty government public institution dependent on the Ministry of Health.

### 2.3. Period of Data Collection

The patient files were reviewed, and their date of birth, geographical origin, sex, age at symptom onset, disorder name, and disorder mode of inheritance were obtained.

The studied cohort comprised patients diagnosed in our center or patients whose IEiM diagnosis was established in other medical units, who were referred to our center for continuity of treatment (hereafter described as “referred”).

Patients were categorized as follows: (1) screened (those with positive NBS results performed at any health facility) and (2) unscreened (those sick children clinically suspected of having IEiM who were hospitalized or ambulatory). In this category, the older unscreened affected siblings were also included.

### 2.4. Case Definition, Confirmation, and Disorder Classification

Diagnostic biochemical confirmatory studies were performed in our center using MS/MS, GC-MS, HPLC, fluorometry, and orotic acid quantification, among other methods, as previously described [29]. Cases were classified as definitive as described by Sontang et al. according to the biochemical results (i.e., the presence of alloisoleucine for maple syrup urine disease (MSUD) cases or elevated Phenylalanine (Phe); diminished tyrosine (Tyr) and a Phe/Tyr ratio > 2 for hyperphenylalaninemia/phenylketonuria (HPA/PKU) cases) [30,31].

The number of patients accessing genetic testing to identify their pathogenic variants and the methodologies used (e.g., Sanger and exome sequencing) were recorded. The funding source for the genetic test (i.e., our governmental institution or other public or private institutions) was also registered. As this study comprised data across 33 years, assuming that advances in genetic testing were different in 1990 compared with the present time, we separated access to genetic testing into three periods: 1990 to 2000, 2001 to 2010, and 2011 to 2022.

### 2.5. Disorder Classification

The different detected IEiM were classified as amino acid metabolism, organic acid, urea cycle, defects of carbohydrate metabolism, lipid metabolism, mitochondrial fatty acid oxidation and related metabolic pathway, vitamin-responsive, and amino acid transport disorders. Their International Classification Disorders Classification 11th Revision of the IEiMs was included. Also, the amenability of each disease to expanded NBS was assessed based on the U.S. recommended uniform newborn screening panel (RUSP) information [31], as well as the existence of currently accepted treatments for each disorder and their availability in Mexico being assessed. NBS critical disorders were also registered. Critical disorders are IEMs that may present acutely in the first weeks of life and require immediate treatment to mitigate morbidity and mortality [32].

### 2.6. Diagnostic Delay, Travel from Habitual Address to Our Reference Center, and Mortality Rates

The diagnostic delay (the time between symptom onset age and the age at diagnosis), the number of other affected siblings in the family, and if death occurred were analyzed. The distance in km from the habitual residence of the patients to the metabolic clinic located in Mexico City was also determined and compared with the proportion of patients with other diseases who routinely come to our institution for medical care (such as cancer, neurosurgery, urology, and nephrology). The distance patients had to travel from their habitual residence place to our institution to have a diagnosis made and obtain treatment was recorded.

Mortality by disease was also assessed for all illnesses. An analysis excluding conditions with the same mortality rate as that observed in the general population, such as HPA/PKU or 3-methylcronyl-CoA carboxylase deficiency (MCC) [33,34], was performed to avoid bias. Kaplan‒Meier survival curves were generated for disorders with the highest mortality rates.

### 2.7. Statistical Methods

Statistical analyses were performed with GraphPad Prism version 9.2. Medians with IQR1 and IQR3 (minimum and maximum) were used for the statistical description of the cohort. Where necessary, a comparison of proportions was performed. Missing data were addressed by adjusting the “*n*” number where applicable.

### 2.8. Ethical Considerations

This study is part of research project number 2022/051, approved by the National Institute of Pediatrics Research, Ethics and Biosafety committees, and was performed according to the guidelines of the Helsinki Declaration.

## 3. Results

Data from 924 patients with IEiM were studied; 445 (48.16%) females and 479 (51.84%) males were analyzed. The data comprised 835 Mexican families, of which 79 had more than one affected child (74 had two affected siblings, 4 had three, and 1 had eight). The older siblings were affected in 39/79 (49.36%) families, and the diagnosis was confirmed as part of the family history. Only one multiple pregnancy (twins) was registered. Consanguinity was documented in 13.5% of the families, and 35/100 (35%) of these families had access to expanded NBS.

We found 40 different IEiMs, and according to the affected metabolic pathway, the most frequent disorders were aminoacidopathies, followed by organic acid disorders (Figure 1).

In Figure 2, a general diagram of the study is presented, showing the number of disorders found and classified by those detectable or not detectable by NBS. Twenty-nine IEiMs were detectable by NBS. The highest number of screened patients were categorized with HPA/PKU (49.3%), followed by galactosemia (GALAC) with 16.8%.

The detected diseases and their respective number of patients are shown in Table 1. The most frequent disorder found was HPA/PKU, which represented 195/924 (21.1%) cases, followed by MMA, which represented 119/924 (12.9%) patients.

An average of 28 new cases arose per year, with the 5 most frequent being HPA/PKU, methylmalonic acidemia (MMA), MSUD, GALAC, and propionic acidemia (PA) (Figure 3).

The patients came from all 32 states of the Mexican Republic. The overall geographical distribution of cases as well as that of the five more frequently diagnosed disorders are shown in Figure 4. Detailed information on the remaining diseases is shown in Appendix A. Remarkably, 80% of the patients had to travel long distances to arrive at our metabolic center; the longest was 2300 km (Table 2).

In the unscreened group, when data on symptom onset and age at diagnosis were available, the diagnostic delay was calculated (*n* = 376 patients), and the results are shown in Figure 5. Homocystinuria (HCY) had the highest diagnostic delay, with a median of 5.2 years, followed by 3-methylglutaconic aciduria (3MGA), with a median diagnostic delay of 2.9 years.

Excluding the referred patients and those whose disease is not detectable with NBS, we analyzed the differences in age at arrival, mortality percentage, and access to genetic tests between screened and unscreened patients diagnosed at our center (Table 3). The median age at arrival at our metabolic center was significantly lower for the screened group (1.5 months) than for the unscreened group (10.1 months), *p* < 0.0001. To avoid bias, we analyzed overall mortality and mortality, excluding diseases with the same mortality rate as the general population (i.e., HPA/PKU and MCC) and those disorders not detectable by NBS. The adjusted percentage of dead patients was higher in the unscreened group (Table 3). Remarkably, the mortality in the unscreened patients was over two-fold higher than that in the screened group (*p* < 0.0001). Kaplan–Meier survival curves of the five disorders with the most increased mortality percentage are shown in Figure 6. It can be observed that a 50% survival probability was at 3 and 6 months for patients with MMA and MSUD, respectively.

Appendix A presents detailed information on patients who underwent genetic testing organized by disease. Only 309/924 (33.4%) patients had access to genetic testing; in 236/309 (76.3%), the genotype was established with Sanger sequencing at our institution, and the remaining 73/309 (23.6%) tests were performed at other private or public institutions with Sanger sequencing or WES. The type of funding used for the genetic test was also considered. The genetic studies conducted at our institution were financed through research protocols; on other occasions, these studies were paid for by the families or a private sponsor, such as a pharmaceutical company. Additionally, access to genetic testing by period was null in the 1996–2000 period, and in the 2001–2010 decade, only 39 patients (4.2% of the analyzed population) underwent genetic testing. From 2011 to 2023, 270 patients (29.2% of the analyzed population) received genetic testing.

The treatments currently described in the literature for the 40 disorders found in this study include nutritional restriction treatment, cofactor supplementation, pharmacological therapy (nitisinone, cysteamine, sapropterin, and carglumic acid), enzyme replacement therapy, and liver or liver–kidney transplantation; most of these treatments are available in Mexico (Appendix A).

## 4. Discussion

Herein, we present the disparities and unmet needs identified in a cohort of Mexican patients with IEiM to achieve an accurate and prompt diagnosis. To our knowledge, this study involves the largest cohort of patients with IEiM described in Mexico in the last 33 years. Regarding the spectrum of IEiM, amino acid disorders and organic acidemias were the most frequent IEiMs (Figure 1), which coincides with previous reports from others worldwide [35,36]. We found that 30/40 (75%) disorders were detectable by expanded NBS and treatment [1,31]. Unfortunately, the most striking disparity was that only 35.4% (293/828) of the patients had access to expanded NBS (Figure 2).

As demonstrated in Figure 4, the geographical distribution of the studied IEiM is variable, showing some clusters. For HPA/PKU, the high prevalence in the central west part of the country has been associated by our group with a founder effect [37,38]. Still, the apparent presence of clusters for other diseases, i.e., PA in the Mexico City area or MSUD in the Pacific area, has yet to be explored and deserves more studies due to the lack of epidemiological data. Additionally, the cluster observed in Tabasco could be due to a pilot expanded NBS program that was performed in that state.

In this study, we observed only seven MCAD cases. This is notably lower from those reported in other populations, mainly Caucasian, where MCAD is one of the most frequent IEiMs [39,40]. Feuchtbam et al., in 2012, reported the birth prevalence of several disorders detectable through NBS by race and ethnicity in California, US, analyzing more than two million newborns [41]. One of the most interesting findings of Feuchtbam et al. is that in the “Hispanic” group, the prevalence of MCAD was notably lower (4.5 per 100,000 newborns) than in Native Americans (13.4 per 100,000 newborns) and Whites (9.4 per 100,000 newborns). Another fact that could explain our findings is that some FAOD, including MCAD, could be undetected due to neonatal or childhood sudden death without a diagnosis. This points out the necessity to perform well-planned NBS pilot programs. Further studies about MCAD birth prevalence are needed in Mexico, especially considering that several authors have demonstrated that expanded NBS for MCAD is cost-effective and could save fatal outcomes in nearly one of seven patients, with low-cost treatment measures (frequently feeding to avoid fasting) [42,43].

Despite Mexico’s NBS program being the first implemented in Latin America [12], it has faced many obstacles and challenges not yet resolved that require urgent modernization. One of the main obstacles NBS programs have met in Mexico is the health system’s complexity, characterized by considerable fragmentation, giving the different population groups unequal access to public resources and other health benefits [44]. Mexico has as many NBS panels as existent health institutions performing screening [16]. Our center is part of the Mexican Ministry of Health, where the mandatory NBS panel includes the detection of only two IEiMs (HPA/PKU and GALAC) and four other diseases (congenital hypothyroidism, congenital adrenal hyperplasia, cystic fibrosis, and glucose-6-phosphate dehydrogenase deficiency) [15]. A possible effect of this mandatory panel could be supported by the fact that in our studied cohort, the two disorders with the highest percentages of screened patients were HPA/PKU (49.3%) and GALAC (16.8%). Nevertheless, the rate of screened patients remains significantly unsatisfactory, given that the targeted coverage for NBS is expected to be 100% [45]. Thus, our patients with IEiM detectable by NBS are frequently detected late.

On the other hand, within the Mexican private healthcare system, the provision of childbirth or cesarean section care is bundled into a comprehensive “package”, which encompasses the expanded NBS and is administered by various healthcare providers. Regrettably, a significant deficiency exists within most private hospitals in our nation, characterized by the absence of systematic follow-up mechanisms for NBS. Consequently, families receiving positive NBS results must request confirmation and treatment at our institution. In other instances, NBS procedures have been conducted in foreign countries, primarily the United States. Owing to the irregular immigration status of these families, they find themselves obliged to return to Mexico, where they seek medical care for their infants.

The benefits of expanded NBS have been profusely documented, with an early diagnosis, early treatment, and a favorable cost–benefit relationship [46,47]. The disparity observed in our population must be corrected, especially considering that the number of patients arriving at our institution has increased over time (Figure 3). Furthermore, due to the lack of expanded NBS for IEiM in Mexico, this work demonstrates that most patients experienced a long diagnostic delay (Figure 5). This is especially relevant for certain IEiM, considered a critical disorder due to its devastating natural history, where the diagnosis must be performed in the first days of life [32].

As it is known, most of the IEiMs are inherited in an autosomal recessive manner, and the risk of an affected product is 25%. In our study, we documented the presence of other affected members in 79 families. This fact emphasizes the importance of obtaining a complete family history in all patients. Performing biochemical or genetic studies on older siblings must be mandatory since affected siblings should receive treatment and follow up, regardless of age. Furthermore, families with an affected IEM child must receive professional genetic counseling, especially considering future pregnancies [29]. The consanguinity rate in this study (13.5%) was noticeably lower than the overall consanguinity meta-analysis estimated at 51.47% by Waters et al., for IEiM worldwide [3], but is higher than the 1–4% previously reported in Mexico [48].

The adjusted mortality observed in this study was significantly higher in the unscreened patients (26.8%) than in the screened patients (14.58%) (Table 3). The highest mortality rate was observed in diseases such as MMA, PA, MSUD, and ornithine transcarbamylase deficiency (OTC) (Figure 6), all of which are detectable by expanded NBS [31]. High mortality for those diseases has also been observed in other cohorts of patients, regardless of the type of diagnosis mode and treatment [49,50,51].

Of the 40 IEiMs detected in the studied patients, thirty-five (87.5%) have a therapy available in Mexico (Appendix A). Since one of the inclusion criteria for disorders to be screened is that they have an available treatment [31], the fact that 87.5% of IEiM disorders found are treatable [52,53,54] gives more support for their inclusion in the NBS mandatory panel in the Mexican Ministry of Health.

Other findings are worth mentioning. The survival probability of patients with some diseases, such as tyrosinemia type 1 (TYR-1), has increased mainly due to the combination of expanded NBS with the availability of early pharmacological therapy (nitisinone) and liver transplantation [24] (Figure 6E).

Another advance is the continuous growth of the overall number of cases at our reference center, with an average of 28 patients per year (Figure 3). The evolution of our technological platform could explain this, mainly the implementation of MS/MS in 2010 and the consolidation of a specialized interdisciplinary team composed of metabolic pediatricians, dietitians, biochemists, geneticists, and other professionals, such as quality-of-life experts. This team exerts joint efforts to increase the knowledge and awareness of health personnel to promote continuous training and diffusion of knowledge about NBS throughout the country [20,21,22,23,26,27,28,55,56,57]. Another reason could be the increase in private laboratories, including the expanded NBS service following the U.S. RUSP panel [12,31]. The observed maximum peak of 56 cases in 2014 could be related to the pilot programs of the NBS for HPA/PKU and GALAC in Mexico and its later inclusion in the national mandatory NBS panel [15]. We observed a critical decline in the number of cases in 2020 and 2021; this could be explained with the COVID-19 pandemic, which affected the number of diagnosed cases reported in other populations of IEM patients [58,59], as well as with the diversion of financial healthcare resources for attending the more significant needs of the COVID-19 pandemic. Another reason could be the decrease in registered births in the 2020 year, where they decreased from 2,092,214 in 2020 to 1,629,211 in 2021 (INEGI, https://www.inegi.org.mx/contenidos/saladeprensa/boletines/2022/NR/NR2021.pdf, consulted on 29 August 2023).

NBS panel selection is a complex public health issue, and the addition of diseases must be performed carefully, considering many factors, such as the benefits, the possible harms, and the economic factors, among others [60,61,62]. In addition, controversies still exist for conditions like 3MCC, which are generally asymptomatic [63], and have even been removed from some NBS programs [64]. MET is also debatable due to its benign clinical course [65]. In our study, we found nine individuals with 3-MCC who did not have NBS; eight of them were siblings of a patient detected by NBS who were asymptomatic. The ninth patient was a 4-year-old child without NBS who presented a metabolic crisis characterized by vomiting, feeding refusal, metabolic acidosis, and seizures, so she was admitted to the emergency department in our institution. In this patient, as part of the metabolic acidosis study protocol, we performed the GCMS study, which showed the characteristic profile of organic acids of 3MCC. We also have another symptomatic 3MCC patient detected by NBS who developed hyperammonemia crises at 6 months, which deserved hospitalization and medical management in the emergency room, with good outcomes. These results show weak evidence supporting whether 3MCC must be deleted or not from NBS programs.

Another concern about the obsolesce of the NBS program of the Mexican Ministry of Health is that while we keep searching for the expansion of the program to detect IEiMs by MS/MS, currently developed countries are already exploring genomic NBS with WES or even WGS techniques [66]. If this gap is not overcome, it will further deepen the inequities in access to NBS between different regions worldwide. Genetic studies are essential for the adequate diagnosis, management (i.e., selection of patients for specific treatment), and prognosis of IEiM patients, especially in the pediatric population and even more so in critically ill patients, allowing personalized medicine approaches [67,68]. These studies are also helpful for the establishment of the prognosis of those cases of MAA where different forms (mut^0^ and mut^-^) of the disease differently impact the patient survival [69,70].

Furthermore, genetic sequencing could be an alternative for diagnosing IEiMs without a specific diagnostic biomarker [71].

Unfortunately, only 309/924 (33.4%) patients had access to molecular studies, and the molecular studies were only performed for 30/40 (75%) disorders; this information is detailed in Appendix A. Most genetic studies of the patients reported herein were performed in the last decade (270/309, 87.4%). This increase in testing coincides with the growing technological advances in DNA sequencing that began in the 1990s and have expanded in recent years [72]. WGS and WES sequencing techniques have demonstrated usefulness [73,74] and advantages over Sanger sequencing. The Sanger technique has limitations in that it is time-consuming and can only analyze one gene at a time [75], but it is still valuable for confirmation of findings on clinical exome sequencing.

At our institution, Sanger sequencing is the only methodology available for genotyping [75]. This is the main reason why we have only performed the genetic studies of six groups of conditions (HPA/PKU [27], TYR-1 [76], GALAC [20], tetrahydrobiopterin defects (BH_4_D) [28], cystinosis (CTNS) [22], and MMA [21]) at our center and only under research protocols, not as permanently available routine tests (Appendix A); thus, molecular studies of the patients were not always performed at the same time of their arrival to our institution. Technological modernization and investment in adequate genotyping methods are needed to fill this gap. A systematized public health program and budget for sequencing large numbers of patients would expedite IEiM diagnoses, especially for those associated with more than one gene (i.e., PA, MSUD, or MMA) or without specific biomarkers for a biochemical diagnosis such as glycogen storage disease (GSD). As other authors have demonstrated, it is necessary to improve access to WES or WGS in this group of patients [77,78,79].

## 5. Conclusions

Our study underscores the critical need for a comprehensive and up-to-date newborn screening (NBS) program in Mexico to address the unmet needs and disparities faced by patients with rare diseases, including IEiMs. Establishing regional medical care centers and implementing public health policies aimed at raising awareness and promoting the early diagnosis and management of these disorders is crucial. Our findings provide compelling evidence that could prove valuable to decision makers overseeing public health initiatives for individuals impacted by IEiMs in middle- and low-income countries.

It is essential to prioritize and allocate resources to develop and implement a well-planned, sustainable, and expanded NBS program for early detection and prompt management of IEiM in Mexico, Latin America, and other middle- and low-income countries. Furthermore, implementing an updated and expanded NBS program can provide valuable insights into the prevalence and distribution of IEiMs and other rare diseases in these countries, which can inform future research and clinical practice. Ultimately, the successful implementation of a comprehensive and sustainable NBS program can significantly impact the health and well-being of individuals and communities affected by these devastating disorders, and it is an urgent public health priority in Mexico and other middle- and low-income countries. Implementing such programs will reduce the burden of IEiM morbidity and mortality on individuals, families, and healthcare systems, leading to better health outcomes and improved quality of life.

It is also important to remark on the necessity of national experts’ consensus who, following the existent scientific evidence and ethical criteria, determine the best NBS panel for Mexico.

Beyond the implementation of comprehensive and sustainable NBS programs is the need for access to permanent treatment for detected patients, along with systematized national algorithms for the massive DNA sequencing of patients to confirm their diseases and to personalize medical care. The establishment of permanent training of health personnel for rare diseases throughout the country is also urgently needed.

## Figures and Tables

**Figure 1 IJNS-09-00059-f001:**
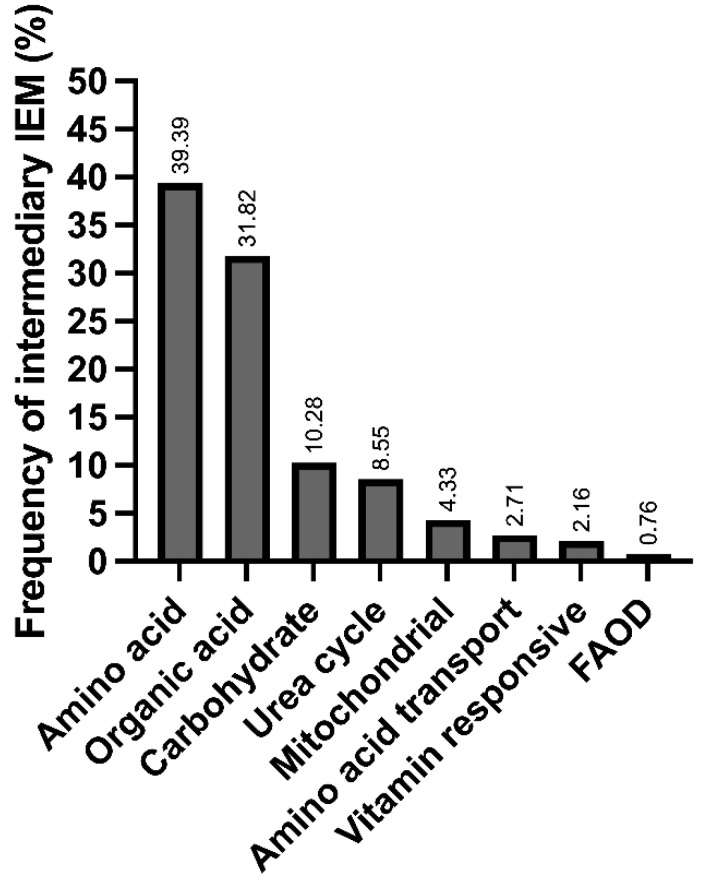
Frequency of inborn errors of intermediary metabolism in the studied population at a Mexican reference center, categorized according to the affected metabolic pathways.

**Figure 2 IJNS-09-00059-f002:**
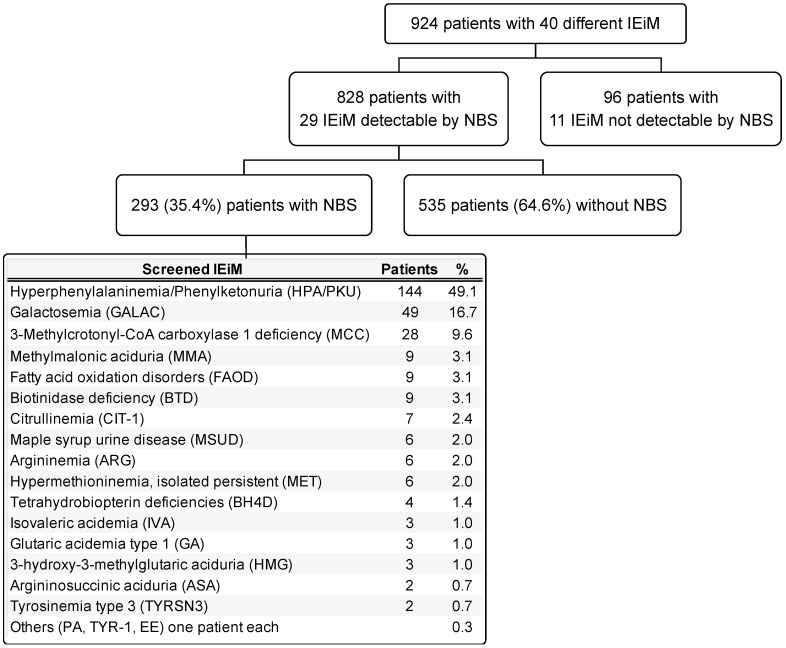
General scheme of the study. The IEiMs found are divided by their detectability or non-detectability through NBS. The group of screened patients is detailed by disease and proportion. Abbreviations. PA, propionic acidemia; TYR-1, tyrosinemia type 1; EE, ethylmalonic encephalopathy.

**Figure 3 IJNS-09-00059-f003:**
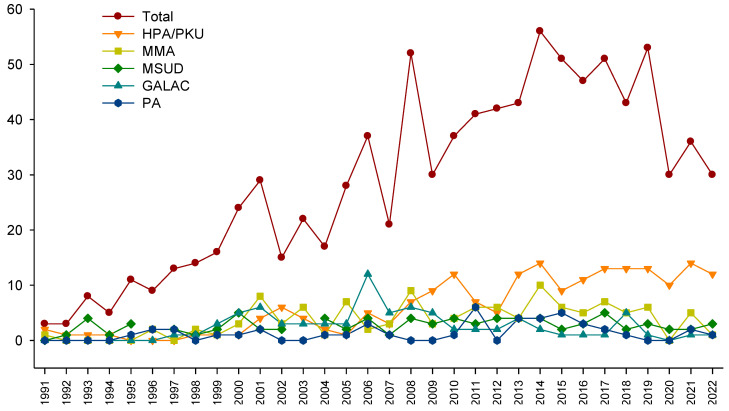
Number of new inborn errors of intermediary metabolism cases detected per year, presented in total (40 disorders), and for the 5 most frequent diseases.

**Figure 4 IJNS-09-00059-f004:**
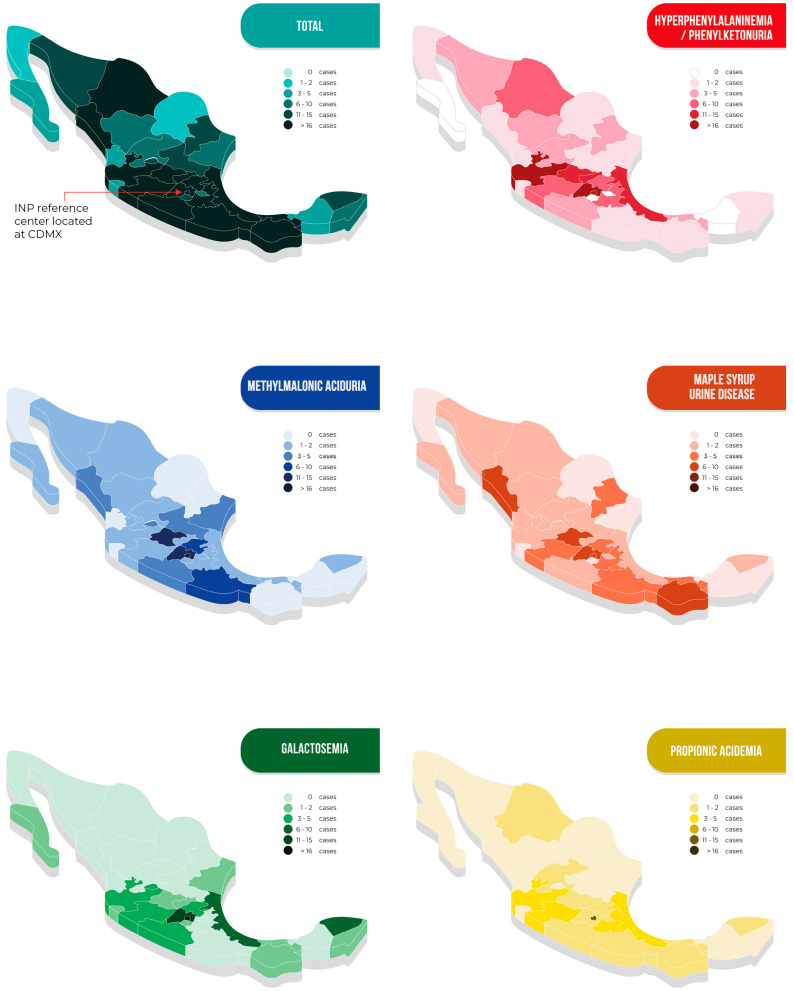
Geographical origin of patients with inborn errors of intermediary metabolism in Mexico, presented in total and for the five more frequently detected disorders.

**Figure 5 IJNS-09-00059-f005:**
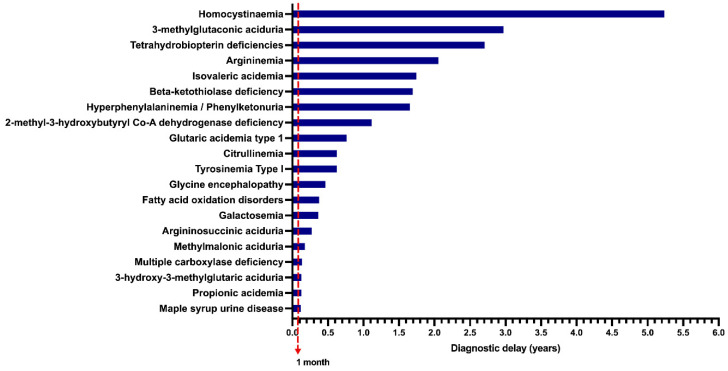
Median of the diagnostic delay of unscreened patients with inborn errors of intermediary metabolism (*n* = 376 patients); 1 month = Limit for early diagnosis time in screened patients.

**Figure 6 IJNS-09-00059-f006:**
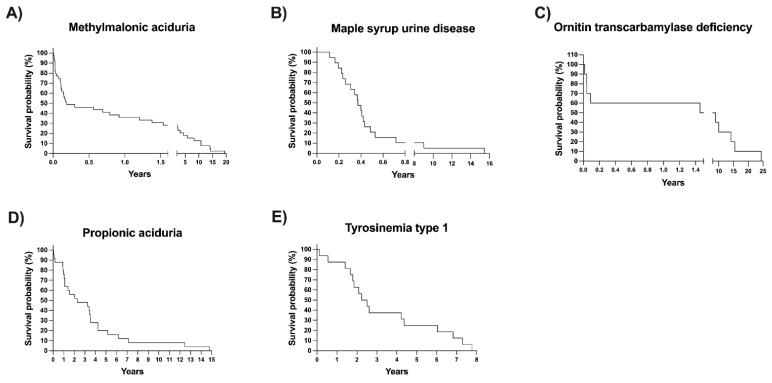
(**A**–**E**) Kaplan‒Meier survival curves of the patients with inborn errors of intermediary metabolism with the highest mortality rates.

**Table 1 IJNS-09-00059-t001:** Inborn errors of intermediary metabolism detected, number of patients, and their classification according to the detection mode (screened, unscreened, and referred).

	Detected in Our Center		Referred from Other Centers **				
	***n* = 727**		***n* = 197**				
	**Screened**	**Unscreened**	**Screened**	**Unscreened**	**Total Number of Patients**	**Gene**	**ICD-11 ***
	** *n* ** ** = 162 (22.3%)**	** *n* ** ** = 565 (77.7%)**	** *n* ** ** = 131 (66.4%)**	** *n* ** ** = 66 (33.5%)**	** *n* ** ** = 924**		
**Detectable by NBS (amino acids, acylcarnitines and succinylacetone)**							
Hyperphenylalaninemia/Phenylketonuria (HPA/PKU)	92	43	52	8	195	*PAH*	5C50.00
Methylmalonic aciduria (MMA)	6	99	3	11	119	*MUT, MMAA, MMAB, MMACHC, MMADHC, MTRR, LMBRD1, MTR, ABCD4, MCEE, SUCLG1, SUCLA2, ACSF3*	5C50.E
Maple syrup urine disease (MSUD)	5	67	1	7	80	*BCKDHB, BCKDHA, DBT*	5C50.D0
Galactosemia (GALAC)	1	23	48	7	79	*GALT*	5C51.4
Propionic acidemia (PA)	1	37		6	44	PCCB, PCCA	5C50.E0
3-Methylcrotonyl-CoA carboxylase 1 deficiency (MCC)	21	9	7		37	MCCC1	5C50.DY
Isovaleric acidemia (IVA)	2	24	1	6	33	*IVD*	5C50.DY
Homocystinaemia (HCY)		24		3	27	*CBS*	5C50.B
Fatty acid oxidation disorders (FAOD)	7	17	2	1	27	*SLC22A5, CPT1, SLC22A5, SLC25A20, SLC25A20, CPT2, ACADVL, ACADM, ACADS, ACAD9, ECHS1, HADHA, HADHB, HADHSC, NADK2, MECR, ETFA, ETFB, ETFDH*	5C52.0
Citrullinemia (CIT-1)	5	15	2	3	25	*ASS1*	5C50.A3
Tyrosinemia Type I (TYR-1)	1	23		1	25	*FAH*	5C50.11
Argininemia (ARG)	4	17	2		23	*ARG1*	5C50.A2
Tetrahydrobiopterin deficiencies (BH4D)	3	14	1	1	19	*PTS, QDPR, PCBD1*	5C50.0Y/5C59
Glutaric acidemia type 1 (GA)	2	12	1	2	17	*GCDH*	5C50.E
3-hydroxy-3-methylglutaric aciduria (HMG)	3	12		1	16	*HMGCL*	5C52.02
Biotinidase deficiency (BTD)	2	1	7	1	11	*BTD*	5C50.E0
Multiple carboxylase deficiency (MCD)		9			9	*HLCS*	5C50.E
Beta-ketothiolase deficiency (BKT)		8			8	*ACAT1*	5C50.DY
Argininosuccinic aciduria (ASA)	1	4	1	1	7	*ASL*	5C50.A0
Hypermethioninemia, isolated persistent (MET)	5		1		6	*MAT1A*	EC50.B
2-methyl-3-hydroxybutyryl Co-A dehydrogenase deficiency (MHDB)		3			3	*HSD17B10*	5C52.01
3-Methylglutaconic aciduria (3MGA)		3			3	*AUH*	5C50.DY
Ethylmalonic encephalopathy (EE)	1	2			3	*ETHE1*	5C50.E
Glycine encephalopathy (GE)		3			3	*AMT, GLDC, GCSH*	5C50.70
Gyrate atrophy of choroid and retina (GACR)		3			3	*OAT*	5C50.9
Succinyl CoA:3-oxoacid CoA transferase deficiency (SCOT)		2			2	OXCT1	5C52.02
Tyrosinemia type 3 (TYRSN3)			2		2	HPD	5C50.1
2-hydroxyglutaric aciduria (D2HGA1)		1			1	*SLC25A1*	5C50.E1
Hyper-β-alaninemia (HBA)				1	1	*Unknown*	5C55.1
**Not detectable to NBS (amino acids, acylcarnitines and succinylacetone)**							
Cystinosis (CTNS)		24			24	*CTNS*	5C60.1
Ornithine transcarbamylase deficiency (OTC)		24			24	*OTC*	5C50.A3
Glycogen storage disease (GSD)		14		1	15	*G6PC, AGL, GBE1, PYGM, PHKA2*	5C51.3
Lactic acidemias and other mitochondrial disorders (MIT)		11		2	13	*PDX1, SURF1, BCS1L*	5C53
Lipoprotein lipase deficiency (LPL)		7			7	*LPL*	5C80.1
N-acetyl aspartic aciduria or Canavan disease (CD)		6			6	*ASPA*	5C50.E1
Alkaptonuria (AKU)		2			2	*HGD*	5C50.10
Glycerol kinase deficiency (GKD)		2			2	*GK*	5C51.1
Glucose-galactose malabsorption (GGM)				1	1	*SLC5A1*	5C61.0
Hartnup disease (HND)				1	1	SLC6A19	5C60.Y
MTHFR deficiency (MTHFRD)				1	1	MTHFR	5C63.1

All the disorders are of autosomal recessive inheritance, except OTC and GSD type IX, which are inherited in an X-linked manner. * International Classification Disorders Classification 11th Revision. ** Initially diagnosed in other medical centers, national or international (mainly USA).

**Table 2 IJNS-09-00059-t002:** Travel distance from the geographical origin to the reference center and percentage of patients who travel from each Mexican state to attend our institution.

	Mexican State	Approximate Travel Distance to the Reference Center at INP (km)	Number (%) IEiM Patients
Short travel distance	CDMX	0	177 (19.2)
	México	63	116 (12.6)
	Morelos	75	12 (1.3)
	Hidalgo	91	46 (5.0)
	Tlaxcala	133	7 (0.8)
	Puebla	147	30 (3.2)
Long travel distance	Querétaro	212	32 (3.5)
	Michoacán	298	34 (3.7)
	Veracruz	308	37 (4.0)
	Guanajuato	330	49 (5.3)
	Guerrero	368	26 (2.9)
	San Luis Potosí	424	13 (1.4)
	Oaxaca	474	31 (3.4)
	Colima	477	3 (0.32)
	Aguascalientes	502	14 (1.5)
	Jalisco	536	49 (5.3)
	Zacatecas	635	10 (1.1)
	Chiapas	705	16 (1.7)
	Nayarit	740	4 (0.4)
	Tabasco	762	68 (7.3)
	Durango	772	6 (0.6)
	Coahuila	818	2 (0.2)
	Campeche	910	3 (0.3)
	Nuevo León	917	12 (1.3)
	Tamaulipas	954	10 (1.1)
	Sinaloa	1218	18 (1.2)
	Yucatán	1323	15 (1.6)
	Chihuahua	1559	20 (2.1)
	Quintana Roo	1621	7 (0.8)
	Baja California Sur	1677	5 (0.5)
	Sonora	1890	11 (1.2)
	Baja California	2300	2 (0.2)
	No data		39 (4.2)

INP: National Institute of Pediatrics by its abbreviation in Spanish.

**Table 3 IJNS-09-00059-t003:** Differences in age at arrival, mortality, and access to genetic tests between screened and unscreened patients with inborn errors of intermediary metabolism.

Total of Patients *n* = 637	Screened *n* = 162	Unscreened *n* = 475
Median age at arrival to our Center in months (IQR) ^a^	1.60 (0.93−2.82) ****	10.10 (1.90−33.37)
Min (days) − Max (months)	1−12.1	1−238.43
General mortality	8/162 (4.94%) ****	123/475 (25.89%)
Adjusted mortality ^b^	7/44 (15.91%) *	123/423 (29.08%)

^a^ IQR: interquartile range. ^b^ Without HPA/PKU, MCC, MET, and those diseases that are not detectable by NBS. Differences between screened and unscreened patients: * *p* = 0.0329, **** *p* < 0.0001.

## Data Availability

The original contributions presented in the study are included in the article; further inquiries can be directed to the corresponding author.

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
