# Peer review of "A Review of Disparities and Unmet Newborn Screening Needs over 33 Years in a Cohort of Mexican Patients with Inborn Errors of Intermediary Metabolism"

_2409-515X, 2023, doi:10.3390/ijns9040059_

Round 1

Reviewer 1 Report

This is a well written paper with good information, particularly for Mexican health policy makers. I am not a fan of using the word “amenable” when you really mean “detected by” and not amenable to mean “not detected by.” To me, amenable means that it could be screened, whether or not it could be detected, but I will leave it up to you. Below are a couple of other suggestions:

Section 2.1 – Substitute “33 year” for “historical” – “…observational study of a 33 year cohort…” for clarity.

Section 2.3 – I think the word “revised” is intended to be “reviewed.’

Section 2.5 -  Please insert U.S. before recommended uniform screening panel for clarity – “… based on the U.S. recommended uniform newborn screening panel (RUSP) information”

Section 3, paragraph 3 – Please reorganize first sentence for clarification. The RUSP is a living recommendation that is updated periodically, so that the numbers can change.  I suggest something like, “Thirty IEiM included on the RUSP were amenable to NBS.”  Also I suggest changing the second sentence to, “The highest number of screened patients were categorized with HPA/PKU (49.3%), followed by galactosemia (GALAC) with 16.8%.”

Conclusions – This section is a little weak. Can you discuss enforcement of the national requirements?  My understanding is that while there are already required conditions, there is no penalty or means of enforcement. This leads to the inequities discussed. Wherever you note an accomplishment or a need, can you please summarize what they are. For example, you note the need for regional centers. Can you briefly describe whether this means one in every political region, one in the northern region, southern region, etc. You note the need for expanded screening. Could you please delineate what this actually means? For example, are you talking about copying the U.S. RUSP or something different (Canada, for example)? You note that compelling evidence has been presented. Could you please summarize? 

Title:  I suggest a change in the title to: “A Review of Disparities and Unmet Newborn Screening Needs Over 33 Years in a Cohort of Mexican Patients with Inborn Errors of Intermediary Metabolism” in order to better describe the study and give credit for its long time span.

No real problem other that was was noted in comments to authors.  The word "amenable" is not my favorite as noted.

Author Response

Manuscript ID:IJNS-2598204 entitled “IDENTIFICATION OF DISPARITIES AND UNMET NEWBORN SCREENING NEEDS IN MEXICAN PATIENTS WITH INBORN ERRORS OF INTERMEDIARY METABOLISM”. As the reviewers suggested, the title was change to “A REVIEW OF DISPARITIES AND UNMET NEWBORN SCREENING NEEDS OVER 33 YEARS IN A COHORT OF MEXICAN PATIENTS WITH INBORN ERRORS OF INTERMEDIARY METABOLISM”.

Dear Reviewer 1:

Please find below the response to your suggestions and comments. Changes made to the manuscript are highlighted in red in the revised version of the manuscript.

This is a well written paper with good information, particularly for Mexican health policy makers. I am not a fan of using the word “amenable” when you really mean “detected by” and not amenable to mean “not detected by.” To me, amenable means that it could be screened, whether or not it could be detected, but I will leave it up to you. Below are a couple of other suggestions:

RESPONSE: Thank you for this suggestion. We have changed the term “amenable” to “detectable” and “not amenable” to “not detectable”.

Section 2.1 – Substitute “33 year” for “historical” – “…observational study of a 33 year cohort…” for clarity.

RESPONSE: Thank you for this suggestion; the “33 years” was replaced by “historical.”

Section 2.3 – I think the word “revised” is intended to be “reviewed.’

RESPONSE: We agree. The word “revised” was changed to “reviewed”

Section 2.5 -  Please insert U.S. before recommended uniform screening panel for clarity – “… based on the U.S. recommended uniform newborn screening panel (RUSP) information”

RESPONSE: We agree that “U.S.” was inserted.

Section 3, paragraph 3 – Please reorganize first sentence for clarification. The RUSP is a living recommendation that is updated periodically, so that the numbers can change.  I suggest something like, “Thirty IEiM included on the RUSP were amenable to NBS.”  Also I suggest changing the second sentence to, “The highest number of screened patients were categorized with HPA/PKU (49.3%), followed by galactosemia (GALAC) with 16.8%.”

RESPONSE: Thank you for this suggestion; the appropriate changes were made.

Conclusions – This section is a little weak. Can you discuss enforcement of the national requirements? My understanding is that while there are already required conditions, there is no penalty or means of enforcement. This leads to the inequities discussed. Wherever you note an accomplishment or a need, can you please summarize what they are. For example, you note the need for regional centers. Can you briefly describe whether this means one in every political region, one in the northern region, southern region, etc. You note the need for expanded screening. Could you please delineate what this actually means? For example, are you talking about copying the U.S. RUSP or something different (Canada, for example)? You note that compelling evidence has been presented. Could you please summarize?

RESPONSE: Thank you for this suggestion. It has enriched the manuscript. The following changes were included in the text:

Even though the basic NBS for six diseases is mandatory in Mexico, there are no penalties for its omissions and errors. The NBS in Mexico is not a priority, and our group has documented a lot of mistakes and iatrogenesis related to it. Unfortunately, clinicians and researchers do not have decision-making power over the NBS national program. Therefore, our only voice is to document and publish what we see in our reference center. Regarding the number or ubication of the regional care centers needed, we are only suggesting the necessity, but their establishment requires the consensus of experts.

In the author's opinion, we would not like to copy or replicate the US NBS expanded panel; instead, we would like to have a sustainable, well-planned NBS program with all the strict indicators of performance, quality, and follow-up.

To accomplish your comments, we change the conclusion as follows:

CONCLUSIONS:

Our study underscores the critical need for a comprehensive and up-to-date newborn screening (NBS) program in Mexico to address the unmet needs and disparities faced by patients with rare diseases, including IEiMs. Establishing regional medical care centers and implementing public health policies aimed at raising awareness and promoting early diagnosis and management of these disorders is crucial. Our findings provide compelling evidence that could prove valuable to decision-makers overseeing public health initiatives for individuals impacted by IEiMs in middle- and low-income countries.

It is essential to prioritize and allocate resources to develop and implement a well-planned, sustainable, and expanded NBS program for early detection and prompt management of IEiM in Mexico, Latin America, and other middle- and low-income countries. Furthermore, implementing an updated and expanded NBS program can provide valuable insights into the prevalence and distribution of IEiMs and other rare diseases in these countries, which can inform future research and clinical practice. Ultimately, the successful implementation of a comprehensive and sustainable NBS program can significantly impact the health and well-being of individuals and communities affected by these devastating disorders, and it is an urgent public health priority in Mexico and other middle- and low-income countries. Implementing such programs will reduce the burden of IEiM morbidity and mortality on individuals, families, and healthcare systems, leading to better health outcomes and improved quality of life.

It is also important to remark on the necessity of national experts' consensus who, following the existent scientific evidence and ethical criteria, determine the best NBS panel for Mexico.

Beyond the implementation of comprehensive and sustainable NBS programs is the need for access to permanent treatment for detected patients, along with systematized national algorithms for the massive DNA sequencing of patients to confirm their diseases and to personalize medical care. The establishment of permanent training of health personnel in rare diseases throughout the country is also urgently needed.

Title:  I suggest a change in the title to: “A Review of Disparities and Unmet Newborn Screening Needs Over 33 Years in a Cohort of Mexican Patients with Inborn Errors of Intermediary Metabolism” in order to better describe the study and give credit for its long time span.

RESPONSE: We agree, the title has been changed.

We thank the reviewer for his/her comments which improved our manuscript. We hope the revisions made are satisfactory.

Reviewer 2 Report

The amount of work to review clinical information on over 900 individuals for this manuscript is appreciated. You have an opportunity to enlighten the screening community about the challenges when a portion of the population receives screening and molecular sequencing without resolution of the inequity in the last decade.

This reviewed had several questions about the population which led to the queries on efficacy of NBS and its sustainability. The challenges of specialized health care delivery across larger geographic distances underscores some of the other comments.

Page 3 (section 2.3 Period of collection, paragraph 3): You need to reconcile your definitions as individuals who were not diagnosed until they were several years of age would no longer be considered infants.  The unscreened category would seem to include sick infants and apparently those who became symptomatic outside of the period of infancy.

Where there are any individuals identified with later presentations of methylmalonic aciduria (including cobalamin C disease), or any of the other classic disorders?  For the OTC patients, does not cohort include any of the later-presenting symptomatic heterozygote females?

Are you willing to estimate what fraction of mitochondrial disease can be detected with the techniques you utilized?  Estimating by lactic acidosis significantly undercounts individuals with complex 1 deficiency as an example.

Do the reported number of individuals with galactosemia include those with the "Duarte2" variant?  As a follow-up, it is there a reason that the frequency of the fatty acid oxidation disorders, which would include medium chain acyl-CoA dehydrogenase deficiency, it is much lower than that of galactosemia when prevalence estimates are approximately 1:23,000 for MCAD alone and 1:40,000 for classic galactosemia?  Both questions go to a limitation of your study, the undetected cases due to neonatal or childhood death without a consideration of an inborn error of metabolism in the differential.  You address the potential for more than 1 affected sibling, but less than 10% of the families had more than 1 affected child.  What fraction of the families had more than 1 birth as you would expect a recurrence risk of 25%?

For comparison purposes a comment should be included about the expected consanguinity rate within Mexico, which has been variously estimated as 1-4%.  Was there a difference between the fractions of families with consanguinity who received screening from those who did not receive screening?

Page 7 (Figure 3): Was there a decrease in birth rates in 2020 and 2021 which partially accounts for the decreased total number of detected individuals with inborn errors of intermediary metabolism?  Where there are gross differences in the detected disorders over the last 3 years from those detected in the previous 10 years?  Were there diversions of healthcare resources away from detection of inborn errors for the greater needs of the COVID-19 pandemic?

In figure 4, the cluster of propionic acidemia cases in CDMX stands out compared to the map for MMA and MSUD. Can this be explained?

Is 3-methylcrotonylcarboxylase deficiency a disease or a biochemical disorder without a clinical phenotype?  Some jurisdictions (worldwide) have excluded it or removed it from MS/MS screening for that reason. You report 9 individuals with that diagnosis who were detected without screening.  Were any symptomatic, or were they all siblings of screened cases?

Were there mut- cases in the MMA population or were all detected cases mut0, which would affect survival? There are data (Horster F et al 2007 Pediatric Research, Horster F et al 2021, J Inherit Metab Dis) for comparison of survival of MMA patients (from screened and unscreened populations). This goes to a trans-national discrepancy in adequacy of care. Were some of the intoxication disorder deaths out-of-hospital/without involvement of appropriate health care professionals? This issue also comes up in figure 6. Are there comparison survival curves for jurisdictions with > 90% inclusion of neonates into a NBS program as comparators? Such data would permit a numeric speculation about the loss of life due to inequities in NBS in Mexico.

Page 10 (Results, penultimate paragraph): The data on genetic testing would be better represented as fraction of individuals in that period. As an example, the ‘39/924’ tested in the 2001-2010 decade is thus higher than 4.2% as the denominator is smaller.

Can you address the cost and sustainability of pharmacological therapy, nutritional restrictional management and cofactor supplementation as a risk factor for mortality in your cohorts? A loss of metabolic control, set up by absence of the best available therapy, leads to higher morbidity and mortality. Sustainability of compliance with therapy is always an issue.

Page 11 (Discussion) Paragraph 5: OTC is not a part of the RUSP panel. Low citrulline alone used for diagnosis is not reliable. Elevated orotic acid alone is also not reliable (Staretz-Chacham O et al 2021J Inherit Metab Dis). This reviewer agrees that theoretically OTC could be detected by NBS at an acceptable false negative level, probably using a combination of analytes and tools such as CLIR (Mayo Foundation of Medical Education and Research). The statement ‘all of which are amenable to expanded NBS’ would be misleading to those unaware of the subtlety and nuance of ‘amenable’ for this case.  

How cost effective is NBS for Mexico? MCAD, if the prevalence is high enough, can justify almost all of a MS/MS approach to NBS because management costs are very low for the disorder and NBS saves nearly 1 in 7 MCAD patients from a fatal outcome (see Iafolla AK et al 1994 and Nennstiel-Ratzel U et al 2005).

Table S1 would be more beneficial if there was a column with the 2020 population data so that the total numbers of recognized IEiM could be calculated on a per 100,000 lives basis (by the reader). Tabasco seems to stand out even when you exclude 3 MCC given a population of about 2.4 million.

Minor:

Page 3 (section 2.5) Line 6: Need to clarify that this is the US recommended panel.

The sentence starting with ‘Consequently’ needs to be rewritten for clarity: ‘…. has as many NBS panels as existent health institutions performing screening’ may be one option.

Page 11 (Discussion paragraph 2):  Change ‘miscellaneous’ to ‘other’ to reduce the risk of a pulmonologist or endocrinologist taking offence.

Consider adding ‘certain’ before ‘IEiM, considered a critical’ as the argument re 3-MCC and a biotinidase disorder would apply to the issue of ‘first days of life’.

Page 12 (Discussion paragraph 9): Reference 56 addresses metabolomics rather than genetic sequencing.  An alternative reference or a change to the sentence is needed.

Page 13 (Discussion paragraph 10): clarify ‘few’ in terms of number performed or breadth of disorders for which such studies were performed.

Sanger sequencing is still used for confirmation of findings on clinical exome sequencing.

Reference 37 (page 15) is incomplete as the source Health Res Policy Sys, volume and pages are missing.

Table S3 (as marked in the supplement) should be table S2

Current table S2 would be more effective if the columns were Disease, Treatments currently available in Mexico, Treatment unavailable in Mexico.  Consider rearranging rows to put the five disorders for which symptomatic treatment is the only option at the bottom of the table.

As table S2 currently reads: Current treatment for PKU/Hyperphenylalaninemia does not include enzyme replacement (of PAH) but enzyme substitution with phenylalanine ammonia lyase (pegvaliase pqpz). 

Minor

Page 2 (section 2.3.  Period of collection, paragraph 1): You did not revise the patient files; you may have intended to say that you reviewed the patient files.  A word change is needed.

Page 3 (Section 2.6) Paragraph 1: English needs to be improved: ‘to have diagnosis’ may be better ‘as to have a diagnosis made and obtain’

Page 9 (Results) Paragraph 2: ‘Withdrawing’ is not the appropriate term. Potential alternatives include ‘Removing’, ‘Excluding’, and the concept of ‘Filtering the data to exclude’

Page 10 (Discussion) paragraph 1: ‘reach’ or ‘achieve’

Page 10 (Discussion) paragraph 2: There is a missing ‘the’ between ‘being’ and ‘first’

Page 12 (Discussion paragraph 8): There may be a missing phrase: ‘diffusion throughout’ might be ‘diffusion of knowledge about NBS throughout’

‘Posterior’ should be changed to ‘later’ or ‘subsequent’.

Table S3 (as marked in the supplement):Tetrahydrobiopterin ‘deficiencies’ would be the correct plural form. Hiper B-alaninemia should be Hyper B-alaninemia.  The same typographical issue exists in current table S2 (next to last row).

Current table S2: (Glycogen storage disease row): typographical error ‘cornstrach’ is ‘cornstarch’ (both columns).  Glycosade needs to be capitalized as it is a trademarked form of heat-modified waxy maize starch. 

Author Response

Manuscript ID:IJNS-2598204 entitled “IDENTIFICATION OF DISPARITIES AND UNMET NEWBORN SCREENING NEEDS IN MEXICAN PATIENTS WITH INBORN ERRORS OF INTERMEDIARY METABOLISM”. As the reviewers suggested, the title was change to “A REVIEW OF DISPARITIES AND UNMET NEWBORN SCREENING NEEDS OVER 33 YEARS IN A COHORT OF MEXICAN PATIENTS WITH INBORN ERRORS OF INTERMEDIARY METABOLISM”.

Dear Reviewer 2:

Please find below the response to your suggestions and comments. Changes made to the manuscript are highlighted in red in the revised version of the manuscript.

The amount of work to review clinical information on over 900 individuals for this manuscript is appreciated. You have an opportunity to enlighten the screening community about the challenges when a portion of the population receives screening and molecular sequencing without resolution of the inequity in the last decade.

This reviewed had several questions about the population which led to the queries on efficacy of NBS and its sustainability. The challenges of specialized health care delivery across larger geographic distances underscores some of the other comments.

Page 3 (section 2.3 Period of collection, paragraph 3): You need to reconcile your definitions as individuals who were not diagnosed until they were several years of age would no longer be considered infants.  The unscreened category would seem to include sick infants and apparently those who became symptomatic outside of the period of infancy.

RESPONSE: We changed the term “infants” for “children” as follows: “2) Unscreened. Those sick children clinically suspected of having IEiM were hospitalized or ambulatory and the affected siblings of positive cases.”

Where there are any individuals identified with later presentations of methylmalonic aciduria (including cobalamin C disease), or any of the other classic disorders?  For the OTC patients, does not cohort include any of the later-presenting symptomatic heterozygote females?

RESPONSE: Yes, we have detected some patients with late-onset forms of methylmalonic aciduria, including cobalamin C diseases, but in this manuscript we did not deeply detail that information, mainly because we have a current protocol of the molecular and clinical spectrum of propionate disorders in Mexico, that we are planning to publish in the next months.

Are you willing to estimate what fraction of mitochondrial disease can be detected with the techniques you utilized?  Estimating by lactic acidosis significantly undercounts individuals with complex 1 deficiency as an example.

RESPONSE: No. unfortunately, mitochondrial diseases is a very complex and large group of disorders, and not all of them have biochemical biomarkers easy to measure. To have specific diagnosis, the molecular studies are needed which we don’t have access in this moment.

Do the reported number of individuals with galactosemia include those with the "Duarte2" variant?  As a follow-up, it is there a reason that the frequency of the fatty acid oxidation disorders, which would include medium chain acyl-CoA dehydrogenase deficiency, it is much lower than that of galactosemia when prevalence estimates are approximately 1:23,000 for MCAD alone and 1:40,000 for classic galactosemia? Both questions go to a limitation of your study, the undetected cases due to neonatal or childhood death without a consideration of an inborn error of metabolism in the differential.

RESPONSE: Thank you for this comment. Yes. We are including the Duarte galactosemia. We don’t know the reason why the number of GALTC cases are higher than MCAD cases. The birth prevalence of most of the IEiM in Mexico is unknown, mainly because there are few expanded screening programs. It is interesting to note that Feuchtbam et al in 2017, reported a big study about the birth prevalence of several disorders, including IEiM, in California, US. The most interesting aspect of Feuchtbam study is that they report the birth prevalence by ethnicity, and in the “Hispanic” group (that comprises a lot of proportion of newborns with Mexican ancestry), the prevalence of MCAD is notably lower (4.5 per 100,000 nbs) than in Native American (13.4 per 100,000 nbs) and Whites (9.4 per 100,000 nbs). Besides, Duarte Galactosemia birth prevalence was also higher (5.6 per 100,000 nbs) than MCAD (4.5 per 100,000 nbs). Of course, your observation that some FAOD including MCAD could be undetected due to neonatal or childhood death without diagnosis, is right, but with cannot confirm this hypothesis with our data, and again these observations point out to the necessity of perform well planned newborn screening pilot programs.

To address your observation, we include in the discussion section the following paragraph:

Interestingly, in this study we observed few MCAD cases compared with other centers [39,40]. This fact remains unknown and deserves deeper studies aimed to elucidate the birth prevalence of IEiM in Mexico. It is interesting to note that Feuchtbam et al in 2012, reported the birth prevalence of several disorders detectable through NBS by race and ethnicity in California, US, analyzing more than two million newborns [41]. One of the most interesting findings of Feuchtbam et al, is that they reported the birth prevalence of the “Hispanic” group, where the prevalence of MCAD was notably lower (4.5 per 100,000 newborns) than in Native Americans (13.4 per 100,000 newborns) and Whites (9.4 per 10.0,000 newborns). Another fact that could explain our findings is that some FAOD including MCAD could be undetected due to neonatal or childhood sudden death without diagnosis. This points out the necessity to perform well planned NBS pilot programs.

We add the following references:

  1. Wilcken B. Fatty acid oxidation disorders: outcome and long-term prognosis. J Inherit Metab Dis. 2010;33(5):501-6. doi: 10.1007/s10545-009-9001-1.
  2. Lüders A, Blankenstein O, Brockow I, Ensenauer R, Lindner M, Schulze A, Nennstiel U; screening laboratories in Germany. Neonatal Screening for Congenital Metabolic and Endocrine Disorders–Results From Germany for the Years 2006–2018. Dtsch Arztebl Int. 2021;118(7):101-108. doi: 10.3238/arztebl.m2021.0009.
  3. Feuchtbaum L, Carter J, Dowray S, Currier RJ, Lorey F. Birth prevalence of disorders detectable through newborn screening by race/ethnicity. Genet Med. 2012;14(11):937-45. doi: 10.1038/gim.2012.76.

You address the potential for more than 1 affected sibling, but less than 10% of the families had more than 1 affected child.  What fraction of the families had more than 1 birth as you would expect a recurrence risk of 25%?

RESPONSE: Thank you for this comment. In the new version of the manuscript, we change the redaction, and we add the row number of families with more than one affected child, as follows:

As it is known, most of the IEiM are inherited in an autosomic recessive manner, and the risk of an affected product is 25%. In our study, we documented the presence of other affected member in 79 families. This fact emphasizes the importance of obtaining a complete family history in all patients. Performing biochemical or genetic studies on older siblings must be mandatory since affected siblings should receive treatment and follow-up, regardless of age. Furthermore, families with an affected IEM child must receive professional genetic counseling, especially considering future pregnancies [29].

For comparison purposes a comment should be included about the expected consanguinity rate within Mexico, which has been variously estimated as 1-4%. Was there a difference between the fractions of families with consanguinity who received screening from those who did not receive screening?

RESPONSE: We are very sorry, because we committed a big mistake in the consanguinity rate expressed in the manuscript. The number of families with consanguinity is 113/835 (13.5%), not 38.9%. In the new version of the manuscript, this mistake was corrected. Regarding your specific question, of the 113 consanguineous families, the number of consanguineous individuals with NBS was 108/192 (56.25%), compared with 235/732 (32.1%) non consanguineous individuals.

In the new version of the manuscript, the following sentences were added in the results section:

“Consanguinity was documented in 13.5% of the families, and 35/100 (35%) of these families had access to expanded NBS”

Regarding the comparison of the previously reported rate of consanguinity in Mexico which varies from 1-4%, we added the following sentence in the Discussion section:

The consanguinity rate in this study (13.5%) was noticeable lower than the overall consanguinity meta-analysis estimated of 51.47% by Waters et al, for IEiM worldwide [3], but is higher than the 1-4% previously reported in Mexico [48].

Page 7 (Figure 3): Was there a decrease in birth rates in 2020 and 2021 which partially accounts for the decreased total number of detected individuals with inborn errors of intermediary metabolism? Where there are gross differences in the detected disorders over the last 3 years from those detected in the previous 10 years? Were there diversions of healthcare resources away from detection of inborn errors for the greater needs of the COVID-19 pandemic?

RESPONSE: Thank you for this comment. Yes. The 3 factors that you mentioned, could explain the observed decrease in the number of cases in the last 3 years. There was a decrease in the birth rate in 2020 in comparison with 2019, but in 2021 the number of registered births increased. This decrease of registered births in 2020 could partially explain the decrease in the total number of detected children with IEiM.

Year

REGISTRED births

2019

2,092,214

2020

1,629,211

2021

1,912,178

The Covid-19 pandemic had an impact on many of the world's newborn screening programs, but in Mexico, it also coincided with a change in government health policy that made changes in the way neonatal screening is financed. Before, in 2018, there was centralized financing for the 6 diseases [15], however, in 2019 the authorities decided to change the financing scheme, making the states pay for their own screening program. This sudden change even caused some state programs to stop. This complex political situation began to be resolved in mid-2020, but in the opinion of the authors, it could have had an influence on the decreased number of cases detected. Unfortunately, this issue has a high political component and we do not have solid evidence to support it in greater depth.

To try to solve your question, we propose the inclusion of the following paragraph: “We observed an important decline in the number of cases in 2020 and 2021; this could be explained by the COVID-19 pandemic, which affected the number of diagnosed cases reported in other populations of IEM patients [51,52], as well as by the diversion of financial healthcare resources for attending the greater needs of the COVID-19 pandemic. Another reason could be the decrease in registered births in 2020 year, where they decreased from 2,092,214 in 2020, to 1,629,211 in 2021 (INEGI, https://www.inegi.org.mx/contenidos/saladeprensa/boletines/2022/NR/NR2021.pdf, consulted on August 29th, 2023).”

In figure 4, the cluster of propionic acidemia cases in CDMX stands out compared to the map for MMA and MSUD. Can this be explained?

RESPONSE:

We don’t have an explanation for this situation, but it deserves further epidemiological studies. We cannot exclude the presence of a possible founder effect, but further studies are needed.

We put the following paragraph in the discussion section, as a limitation of the study:

As demonstrated in Figure 4, the geographical distribution of the studied IEiM is variable, showing some clusters. For HPA/PKU the high prevalence in the central west part of the country has been associated by our group to a founder effect [37,38], but the apparent presence of clusters for other diseases (i.e., PA in Mexico City area, or MSUD in the Pacific area has not been explored and deserves more studies, due to the lack of epidemiological studies.

Is 3-methylcrotonylcarboxylase deficiency a disease or a biochemical disorder without a clinical phenotype?  Some jurisdictions (worldwide) have excluded it or removed it from MS/MS screening for that reason. You report 9 individuals with that diagnosis who were detected without screening. Were any symptomatic, or were they all siblings of screened cases?

RESPONSE: Thank you for this observation. Certainly, 3MCC newborn screening is controversial and some programs are considering eliminating this disorder from the NBS panel.

From these 9 individuals who didn’t have NBS, 8 were siblings from a patient detected by NBS, who were asymptomatic. The 9th patient was a 4-year-old child without neonatal screening, who presented a metabolic crisis characterized by vomiting, feeding refusal, metabolic acidosis, and seizures so she was admitted to the emergency department in our institution. As part of the metabolic acidosis study protocol, we performed the GCMS study which showed the characteristic profile of organic acids. In our historical cohort, we have another symptomatic 3MCC case who was detected by NBS but at the age of six months developed an hyperammonemia crises, that deserved hospitalization and medical management in the emergency room. Therefore, we have weak evidence that supports if 3MCC must be deleted from NBS programs.

To satisfy your comment, we added the following paragraph to the discussion section:

NBS panel selection is a complex public health issue, and the addition of diseases must be done carefully, considering many factors, such as the benefits, the possible harms, and the economic factors, among others [60, 61, 62]. Besides, controversies still exist for conditions like 3MCC, which are generally asymptomatic and have even been removed from some NBS programs [64]. In our study, we found nine individuals with 3-MCC who did not have NBS; eight of them were siblings of a patient detected by NBS who were asymptomatic. The ninth patient was a 4-year-old child without NBS who presented a metabolic crisis characterized by vomiting, feeding refusal, metabolic acidosis, and seizures, so she was admitted to the emergency department in our institution. In this patient, as part of the metabolic acidosis study protocol, we performed the GCMS study, which showed the characteristic profile of organic acids of 3MCC. We also have another symptomatic 3MCC case detected by NBS but developed hyperammonemia crises at six months, which deserved hospitalization and medical management in the emergency room, with good outcome. These results show weak evidence supporting whether 3MCC must be deleted or not from NBS programs.

We also add the following references:

  1. Gray JA, Patnick J, Blanks RG. Maximising benefit and minimising harm of screening. BMJ. 2008 Mar 1;336(7642):480-3. doi: 10.1136/bmj.39470.643218.94.
  2. Odenwald B, Brockow I, Hanauer M, Lüders A, Nennstiel U. Is Our Newborn Screening Working Well? A Literature Review of Quality Requirements for Newborn Blood Spot Screening (NBS) Infrastructure and Procedures. Int J Neonatal Screen. 2023;9(3):35. doi: 10.3390/ijns9030035.
  3. Maier EM, Mütze U, Janzen N, Steuerwald U, Nennstiel U, Odenwald B, Schuhmann E, Lotz-Havla AS, Weiss KJ, Hammersen J, Weigel C, Thimm E, Grünert SC, Hennermann JB, Freisinger P, Krämer J, Das AM, Illsinger S, Gramer G, Fang-Hoffmann J, Garbade SF, Okun JG, Hoffmann GF, Kölker S, Röschinger W. Collaborative evaluation study on 18 candidate diseases for newborn screening in 1.77 million samples. J Inherit Metab Dis. 2023 Aug 21. doi: 10.1002/jimd.12671.

Were there mut- cases in the MMA population or were all detected cases mut0, which would affect survival? There are data (Horster F et al 2007 Pediatric Research, Horster F et al 2021, J Inherit Metab Dis) for comparison of survival of MMA patients (from screened and unscreened populations). This goes to a trans-national discrepancy in adequacy of care. Were some of the intoxication disorder deaths out-of-hospital/without involvement of appropriate health care professionals? This issue also comes up in figure 6. Are there comparison survival curves for jurisdictions with > 90% inclusion of neonates into a NBS program as comparators? Such data would permit a numeric speculation about the loss of life due to inequities in NBS in Mexico.

RESPONSE: Thank you. As described elsewhere, the outcome of mut0 patients is in general, worse, including fatal outcomes. Unfortunately, we don’t have molecular studies of all our MMA patients, thus, we can’t discriminate between them. Currently, we are working in a project about the molecular spectrum of MMA in Mexico, and we recently receive the fundings for that purpose. Regarding the possible transnational discrepancy, and the intoxication disorders deaths, although your comment is very cleaver, unfortunately we do not have access to the information of the NBS programs of our country. We only have our institutional data.

To satisfy your comment, in the discussion section, the following paragraph was added:

“…these studies are also helpful for the establishment of the prognosis of those cases of MAA where different forms (mut0 and mut-) of the disease differently impact the survival patient [69, 70].”

We also added the suggested references:

  1. Hörster F, Tuncel AT, Gleich F, Plessl T, Froese SD, Garbade SF, Kölker S, Baumgartner MR; Additional Contributors from E-IMD. Delineating the clinical spectrum of isolated methylmalonic acidurias: cblA and mut. J Inherit Metab Dis. 2021;44(1):193-214. doi: 10.1002/jimd.12297.
  2. Hörster F, Baumgartner MR, Viardot C, Suormala T, Burgard P, Fowler B, Hoffmann GF, Garbade SF, Kölker S, Baumgartner ER. Long-term outcome in methylmalonic acidurias is influenced by the underlying defect (mut0, mut-, cblA, cblB). Pediatr Res. 2007;62(2):225-30. doi: 10.1203/PDR.0b013e3180a0325f.

Page 10 (Results, penultimate paragraph): The data on genetic testing would be better represented as fraction of individuals in that period. As an example, the ‘39/924’ tested in the 2001-2010 decade is thus higher than 4.2% as the denominator is smaller.

RESPONSE: Thank you for this observation. The molecular studies of the patients were not always performed at the same time of their arrival to our institution. This is mainly due to the absence of budget for sequencing services in our institution, and because, as we mentioned in the manuscript we obtain our molecular results through research protocols, however we did not clearly express this situation. To clarify, we rephrase as follows:

“This is the main reason why we have only performed the genetic studies of six groups of conditions (HPA/PKU [47], TYR-1 [61], GALAC [20], tetrahydrobiopterin defects (BH4D) [28], cystinosis (CTNS) [22], and MMA [21]) at our center and only under research protocols, not as permanently available routine tests (Table S2), thus, molecular studies of the patients were not always performed at the same time of their arrival to our institution.”

Can you address the cost and sustainability of pharmacological therapy, nutritional restrictional management and cofactor supplementation as a risk factor for mortality in your cohorts? A loss of metabolic control, set up by absence of the best available therapy, leads to higher morbidity and mortality. Sustainability of compliance with therapy is always an issue.

RESPONSE: Thank you for this important comment. We do not have enough elements to address the cost and sustainability of the therapies in Mexico: In our reference center, that is a governmental public national health institute, fortunately we have budget for the main therapies i.e, metabolic formulas, orphan drugs such as nitisinone, sapropterine, cysteamine, and triheptanoine. Treatments and consultations are free of charge for the patients, but travel and accommodation expenses to visit our center must be paid by the families. Therefore, this represents a great financial burden for the affected families and strategies must be implemented to solve this problem. We consider that integral and proper medical care of patients with IEiM, especially for those detected by the different NBS programs, must include a well-planned and permanent budget, not only in childhood, but also in adulthood and throughout life. We have a lot of work to do, to raise awareness and improve the current situation of Mexican patients with IEiM and other rare diseases.

We consider that the new version of the conclusion, could satisfy your comment.

Page 11 (Discussion) Paragraph 5: OTC is not a part of the RUSP panel. Low citrulline alone used for diagnosis is not reliable. Elevated orotic acid alone is also not reliable (Staretz-Chacham O et al 2021J Inherit Metab Dis). This reviewer agrees that theoretically OTC could be detected by NBS at an acceptable false negative level, probably using a combination of analytes and tools such as CLIR (Mayo Foundation of Medical Education and Research). The statement ‘all of which are amenable to expanded NBS’ would be misleading to those unaware of the subtlety and nuance of ‘amenable’ for this case. 

RESPONSE: Thank you for your observation; we agree. OTC is not part of the RUSP and could not be detected using the CLIR tools (Hall et al, 2022). In our study, none of the 24 OTC patients were detected by NBS; all of them were detected after symptoms onset and quantifying urinary orotic acid. To satisfy your question, we relocated OTC as a non-detectable disease by NBS (previously named “not amenable”) (Table 1), and as a consequence, we corrected all the numbers throughout the entire manuscript, and we rebuilt Figure 2.

Hall, P. L., Wittenauer, A. L., & Wilcox, W. R. (2022, June). Proximal urea cycle defects are challenging to detect with newborn screening: Results of a prospective pilot study using post-analytical tools. In American Journal of Medical Genetics Part C: Seminars in Medical Genetics (Vol. 190, No. 2, pp. 178-186). Hoboken, USA: John Wiley & Sons, Inc..

How cost effective is NBS for Mexico? MCAD, if the prevalence is high enough, can justify almost all of a MS/MS approach to NBS because management costs are very low for the disorder and NBS saves nearly 1 in 7 MCAD patients from a fatal outcome (see Iafolla AK et al 1994 and Nennstiel-Ratzel U et al 2005).

RESPONSE:

There are no published studies about the possible cost of an expanded NBS in Mexico. We agree that the benefits of MCAD screening would be enough to propose its inclusion in the panel, but the decision of which is the best NBS panel for our country, in the author's opinion, must be made by an expert group consensus, with the participation of the National Health authorities, public health experts, health professionals, and other actors involved.

We also include the suggested references and the following paragraph:

Further studies about MCAD birth prevalence are needed in Mexico, especially considering that several authors have demonstrated that expanded NBS for MCAD is cost-effective and could save fatal outcomes in nearly one in seven patients, with low-cost treatment measures (frequently feeding to avoid fasting) [42,43].

  1. Iafolla AK, Thompson RJ Jr, Roe CR. Medium-chain acyl-coenzyme A dehydrogenase deficiency: clinical course in 120 affected children. J Pediatr. 1994;124(3):409-15. doi: 10.1016/s0022-3476(94)70363-9.
  2. Nennstiel-Ratzel U, Arenz S, Maier EM, Knerr I, Baumkötter J, Röschinger W, Liebl B, Hadorn HB, Roscher AA, von Kries R. Reduced incidence of severe metabolic crisis or death in children with medium chain acyl-CoA dehydrogenase deficiency homozygous for c.985A>G identified by neonatal screening. Mol Genet Metab. 2005;85(2):157-9. doi: 10.1016/j.ymgme.2004.12.010.

Table S1 would be more beneficial if there was a column with the 2020 population data so that the total numbers of recognized IEiM could be calculated on a per 100,000 lives basis (by the reader). Tabasco seems to stand out even when you exclude 3 MCC given a population of about 2.4 million.

RESPONSE: Done. We added the population of each Mexican state in Table S1. In the case of Tabasco, the higher observed number of patients could be a bias, since a pilot expanded NBS program was run in that state, but due to budged restrictions, it was canceled.

To satisfy this comment, the following was added to the discussion section:

“As demonstrated in Figure 4, the geographical distribution of the studied IEiM is variable, showing some clusters. For HPA/PKU, the high prevalence in the central west part of the country has been associated by our group with a founder effect [37,38]. Still, the apparent presence of clusters for other diseases i.e., PA in the Mexico City area, or MSUD in the Pacific area has not been explored and deserves more studies, due to the lack of epidemiological studies. Additionally, some observed clusters could also be due to pilot-expanded NBS programs performed in some states. “

Minor:

Page 3 (section 2.5) Line 6: Need to clarify that this is the US recommended panel.

RESPONSE: Done.

The sentence starting with ‘Consequently’ needs to be rewritten for clarity: ‘…. has as many NBS panels as existent health institutions performing screening’ may be one option.

RESPONSE: Thank you, we change as follows: Mexico has as many NBS panels as existent health institutions performing screening [16].

Page 11 (Discussion paragraph 2):  Change ‘miscellaneous’ to ‘other’ to reduce the risk of a pulmonologist or endocrinologist taking offence.

RESPONSE: Thank you for this comment. We change the term, as you suggested.

Consider adding ‘certain’ before ‘IEiM, considered a critical’ as the argument re 3-MCC and a biotinidase disorder would apply to the issue of ‘first days of life’.

RESPONSE:Done.

Page 12 (Discussion paragraph 9): Reference 56 addresses metabolomics rather than genetic sequencing.  An alternative reference or a change to the sentence is needed.

RESPONSE:

Thank you, we change the reference 56: Chen T, Fan C, Huang Y, Feng J, Zhang Y, Miao J, et al. Genomic Sequencing as a First-Tier Screening Test and Outcomes of Newborn Screening. JAMA Netw Open. 2023;6(9):e2331162. doi: 10.1001/jamanetworkopen.2023.31162.

Page 13 (Discussion paragraph 10): clarify ‘few’ in terms of number performed or breadth of disorders for which such studies were performed.

RESPONSE: To clarify the term “few”, we added the following sentence:

“It is interesting to note that only 309/924 (33.4%) patients had access to molecular study, and the molecular studies only were performed for 30/40 (75%) disorders, this information is detailed in Table S2.”

Sanger sequencing is still used for confirmation of findings on clinical exome sequencing.

RESPONSE: Yes, we agree with this comment. We added the following phrase: The Sanger technique has limitations in that it is time-consuming and can only analyze one gene at once [60], but it is still valuable for confirmation of findings on clinical exome sequencing.

Reference 37 (page 15) is incomplete as the source Health Res Policy Sys, volume and pages are missing.

RESPONSE: We apologize for this mistake. In the new version of the manuscript reference 37 has been corrected as follows: Gómez-Dantés O, Flamand L, Cerecero-García D, Morales-Vazquez M, Serván-Mori E. Origin, impacts, and potential solutions to the fragmentation of the Mexican health system: a consultation with key actors. Health Res Policy Syst. 2023;21(1):80. doi: 10.1186/s12961-023-01025-2.

Table S3 (as marked in the supplement) should be table S2

RESPONSE: We apologize for this mistake. The correction was made in the supplementary material.

Current table S2 would be more effective if the columns were Disease, Treatments currently available in Mexico, Treatment unavailable in Mexico.  Consider rearranging rows to put the five disorders for which symptomatic treatment is the only option at the bottom of the table.

RESPONSE: Thank you for this suggestion. We change table S2, with the following columns: 1) Disease, 2) Current conventional pharmacological treatment worldwide (excluding gene therapy and other experimental therapies) and 3) Treatments currently available in Mexico.

As table S2 currently reads: Current treatment for PKU/Hyperphenylalaninemia does not include enzyme replacement (of PAH) but enzyme substitution with phenylalanine ammonia lyase (pegvaliase pqpz).

RESPONSE: Thank you, we change for enzyme substitution (pegvaliase).

Comments on the Quality of English Language

Minor

Page 2 (section 2.3.  Period of collection, paragraph 1): You did not revise the patient files; you may have intended to say that you reviewed the patient files.  A word change is needed.

RESPONSE: Sorry for this mistake, we corrected it.

Page 3 (Section 2.6) Paragraph 1: English needs to be improved: ‘to have diagnosis’ may be better ‘as to have a diagnosis made and obtain’.

RESPONSE: The suggested change has been done.

Page 9 (Results) Paragraph 2: ‘Withdrawing’ is not the appropriate term. Potential alternatives include ‘Removing’, ‘Excluding’, and the concept of ‘Filtering the data to exclude’

RESPONSE: Thank you for this observation, we exchanged the word “withdrawing” by “excluding”.

Page 10 (Discussion) paragraph 1: ‘reach’ or ‘achieve’

RESPONSE: Thank you, we change “reach” for “achieve”.

Page 10 (Discussion) paragraph 2: There is a missing ‘the’ between ‘being’ and ‘first’

RESPONSE: Sorry for this mistake, we already corrected it.

Page 12 (Discussion paragraph 8): There may be a missing phrase: ‘diffusion throughout’ might be ‘diffusion of knowledge about NBS throughout’

RESPONSE: Sorry for this mistake, we already corrected it.

Posterior’ should be changed to ‘later’ or ‘subsequent’.

RESPONSE: Thank you, we changed “posterior” for “later”.

Table S3 (as marked in the supplement):Tetrahydrobiopterin ‘deficiencies’ would be the correct plural form. Hiper B-alaninemia should be Hyper B-alaninemia.  The same typographical issue exists in current Current table S2: (Glycogen storage disease row): typographical error ‘cornstrach’ is ‘cornstarch’ (both columns).  Glycosade needs to be capitalized as it is a trademarked form of heat-modified waxy maize starch.

RESPONSE: We apologize for these mistakes, in the new version of the manuscript the mistakes were corrected.

We thank the reviewer for his/her comments which improved our manuscript. We hope the revisions made are satisfactory.

Reviewer 3 Report

The authors provide a retrospective observational study of 924 patients with IEiM. Of them, 75% of the diseases can be amenable to NBS, but only 34.3%  screened and unscreened group with higher mortaility and morbidity. Consanguinity was high 38.9% of the families. The data justified for a territory-wide mandatory expanded NBS for Mexico. 

1. Part 2.6, pls clarify why HPA/PKU would have same mortality rate as that observed in general population. In fact, morbidity should also be considered for conditions in NBS. "An analysis excluding conditions with the same mortality rate as that observed in the general population, such as HPA/PKU or 3-methylcronyl-CoA carboxylase deficiency (MCC) [33,34],"

2. Figure 2, is Citrullinemia (CIT) type I or type II? In table I, should be type I then, as ASS1 gene. Please specify. 

3. Figure 2, Hypermethioninemia, isolated persistent (MET) due to MAT1A gene mutations. It can be a benign course to most patients especially AD. Most patients in table 1 indeed are screened likely due to the incidental finding of high isolated methionine in NBS. Hence, the author could consider excluding this condition from the data or explain in the discussion a bit.  Same as 3MCCC.

4. Table 1 subtitle "Not amenable to NBS by MSMS + Fluorometry", actually it depends on what analytes are measured by the MSMS, it can pick up some of the conditions listed. Suggest to revise the subtitle with more specific terms instead. 

5. Among the screened cases, what's the year period that NBS started to be available? Can the authors give us more background information in how these babies received expanded NBS in Mexico? Private hospitals? more well educated parents etc?

Author Response

Manuscript ID:IJNS-2598204 entitled “IDENTIFICATION OF DISPARITIES AND UNMET NEWBORN SCREENING NEEDS IN MEXICAN PATIENTS WITH INBORN ERRORS OF INTERMEDIARY METABOLISM”. As the reviewers suggested, the title was change to “A REVIEW OF DISPARITIES AND UNMET NEWBORN SCREENING NEEDS OVER 33 YEARS IN A COHORT OF MEXICAN PATIENTS WITH INBORN ERRORS OF INTERMEDIARY METABOLISM”.

Dear Reviewer 3:

Please find below the response to your suggestions and comments. Changes made to the manuscript are highlighted in red in the revised version of the manuscript.

Comments and Suggestions for Authors

The authors provide a retrospective observational study of 924 patients with IEiM. Of them, 75% of the diseases can be amenable to NBS, but only 34.3%  screened and unscreened group with higher mortaility and morbidity. Consanguinity was high 38.9% of the families. The data justified for a territory-wide mandatory expanded NBS for Mexico.

RESPONSE: Thank you for your comments. In order to clarify, and as requested by other reviewers, the word “amenable” was replaced by “detectable”, and “not amenable” by “not detectable”.

  1. Part 2.6, pls clarify why HPA/PKU would have same mortality rate as that observed in general population. In fact, morbidity should also be considered for conditions in NBS. "An analysis excluding conditions with the same mortality rate as that observed in the general population, such as HPA/PKU or 3-methylcronyl-CoA carboxylase deficiency (MCC) [33,34],"

RESPONSE:We exclude HPA/PKU based in the studies of Chen, that compare the comorbidities and mortality of HPA/PKU patients vs. general population, concluding that is similar in both populations (reference 33). In the case of 3MCC, it has been described as a condition not clinically significant (Feuchtbaum L, et al 2018.), and nowadays, its inclusion in the NBS panels is controversial. Additionally in Table 3, we also exclude MAT1 cases, as this condition has been also considered as not clinically significant (Chadwick S, 2014).

As other reviewer had a similar query, we added to the new version of the manuscript the following paragraph in the discussion section, and we added the applicable references.

“Besides, controversies still exist for conditions like 3MCC, which are generally asymptomatic [63], and have even been removed from some NBS programs [64]. MAT-1 deficiency is also debatable, due to the mainly benign clinical course [65].”

  1. Feuchtbaum L, Yang J, Currier R. Follow-up status during the first 5 years of life for metabolic disorders on the federal Recommended Uniform Screening Panel. Genet Med. 2018;20(8):831-839. doi: 10.1038/gim.2017.199.
  2. Wilcken B. 3-Methylcrotonyl-CoA carboxylase deficiency: to screen or not to screen? J Inherit Metab Dis. 2016;39(2):171-2. doi: 10.1007/s10545-015-9906-9. Epub 2015 Dec 11. PMID: 26660660.
  3. Chadwick S, Fitzgerald K, Weiss B, Ficicioglu C. Thirteen Patients with MAT1A Mutations Detected Through Newborn Screening: 13 Years' Experience. JIMD Rep. 2014;14:71-6. doi: 10.1007/8904_2013_286.
  4. Figure 2, is Citrullinemia (CIT) type I or type II? In table I, should be type I then, as ASS1 gene. Please specify.

RESPONSE: Thank you for this observation. In the new version of the manuscript, this omission has been solved. We add “type I” after the word citrullinemia.

  1. Figure 2, Hypermethioninemia, isolated persistent (MET) due to MAT1A gene mutations. It can be a benign course to most patients especially AD. Most patients in table 1 indeed are screened likely due to the incidental finding of high isolated methionine in NBS. Hence, the author could consider excluding this condition from the data or explain in the discussion a bit. Same as 3MCCC.

RESPONSE: Thank you for this important comment. MET and 3MCC newborn screening is controversial. According to your suggestion, we excluded MET and 3MCC from the mortality analyses (“adjusted mortality” in Table 3), and in the discussion section, we added the following paragraph:

“Besides, controversies still exist for conditions like 3MCC, which are generally asymptomatic [63], and have even been removed from some NBS programs [64] MET is also debatable due to the mainly benign clinical course [65].”

  1. Table 1 subtitle "Not amenable to NBS by MSMS + Fluorometry", actually it depends on what analytes are measured by the MSMS, it can pick up some of the conditions listed. Suggest to revise the subtitle with more specific terms instead.

RESPONSE:

Thank you for this comment. We agree, to clarify, the subtitle in Table 1 was changed for: “Detectable by NBS (amino acids, acylcarnitines and succinylacetone)” and “Not detectable by NBS (amino acids, acylcarnitines and succinylacetone)”. Additionally, as other reviewers suggested a similar query, in the new version of the manuscript, we changed the term “amenable” to “detectable”, to clarify the idea.

  1. Among the screened cases, what's the year period that NBS started to be available? Can the authors give us more background information in how these babies received expanded NBS in Mexico? Private hospitals? more well educated parents etc?

RESPONSE:

Thank you for this important comment. Expanded NBS began in Mexico in 2000 in some private hospitals (Velázquez 2000). Besides, some Mexican states such as Tabasco, Nuevo León, and Yucatán have implemented pilot expanded NBS studies. On the other hand, most of the patients who arrived at our institution with expanded NBS came from private hospitals. In these hospitals, childbirth or cesarean section care is paid as a “package”, which includes the expanded NBS carried out by different providers. Unfortunately, most of these hospitals in our country lack follow-up systems for NBS, and the families with positive NBS results seek to be confirmed and treated in our institution. In other cases, NBS was performed in another country, mainly the USA, and due to the irregular migratory status of the families, they must return to Mexico and search for medical care and follow-up for their babies.

To satisfy your question, we added the following paragraph to the new version of the manuscript:

“Within the Mexican private healthcare system, the provision of childbirth or cesarean section care is bundled into a comprehensive "package," which encompasses the expanded NBS and is administered by various healthcare providers. Regrettably, a significant deficiency exists within most private hospitals in our nation, characterized by the absence of systematic follow-up mechanisms for NBS. Consequently, families receiving positive NBS results must seek confirmation and treatment at our institution. In other instances, NBS procedures have been conducted in foreign countries, primarily the United States. Owing to the irregular immigration status of these families, they find themselves obliged to return to Mexico, where they seek medical care for their infants.”

We thank the reviewer for his/her comments which improved our manuscript. We hope the revisions made are satisfactory.

Round 2

Reviewer 2 Report

Thank you for your detailed explanations and responses to the comments of the reviewer of the prior draft.  You significantly strengthened the manuscript by the inclusion of new material, especially in the discussion and conclusions sections and the new references.

In the next to last paragraph of the results, the presentation 39/924 leads the casual reader to an incorrect conclusion.  This could be solved by removing the fraction and just stating 39 patients (4.2% of the analyzed population)… and similarly in the next sentence use parallel construction.  Change the second sentence in paragraph 15 of the discussion to (270/309, 87.4%) to emphasize your point about the growth of molecular genetic evaluations.

Methods: Section 2.3: Clarification is needed for the last sentence: 'Those sick children clinically suspected of having IEiM who were hospitalized or ambulatory'… The second portion of the sentence also needs clarification as it’s not clear in English if this is an and/or situation.  Are you referring to children who had a sibling that had a positive screen for an inborn error of metabolism?

Section 2.6: The word ‘registered’ is used multiple times within this section.  Perhaps on line four of the first paragraph, you could use the word ‘determined’.  Likewise in the second paragraph, ‘assessed’ might be substituted for ‘registered and analyzed’ in the first sentence.

Section 3: In the first line, you are not analyzing the individuals, but you are analyzing results from those individuals.  Simple clarification would be after the ’;‘ to include ‘data from’

Section 3: Paragraph 7: Is ‘whether’ the word you intended or did you mean ‘if’ or ‘when’?  This reviewer would tend to use ‘when’ in that situation.

Discussion: First paragraph, line 3: Typographical error: ‘his’ should be ‘this’

Discussion: Paragraph 12: consider changing ‘but’ to ‘who’

Discussion Paragraph 13: Make the inserted new material ’these studies are also helpful..’ a new sentence

Discussion: Paragraph 15: For clarity in English, on the next the last line change ‘once’ to ‘a time’

Conclusion: Paragraph 2 line one: Need an ‘a’ before national.

Author Response

Dear Reviewer 2: Thank you for your prompt response. Please find below the answers point by point to your comments of this second round. Also, we have attached the revised version of the manuscript, that includes in red the changes made.

In the next to last paragraph of the results, the presentation 39/924 leads the casual reader to an incorrect conclusion.  This could be solved by removing the fraction and just stating 39 patients (4.2% of the analyzed population)… and similarly in the next sentence use parallel construction.  Change the second sentence in paragraph 15 of the discussion to (270/309, 87.4%) to emphasize your point about the growth of molecular genetic evaluations.

Response: We agree. The changes were made in the corresponding sections.

Methods: Section 2.3: Clarification is needed for the last sentence: 'Those sick children clinically suspected of having IEiM who were hospitalized or ambulatory'… The second portion of the sentence also needs clarification as it’s not clear in English if this is an and/or situation.  Are you referring to children who had a sibling that had a positive screen for an inborn error of metabolism?

Response: We feel sorry for these imprecisions. The section was corrected as follows:

“The studied cohort comprised patients diagnosed in our center or patients whose IEiM diagnosis was established in other medical units, who were referred to our center for continuity of treatment (hereafter described as “referred”).

Patients were categorized as follows: 1) Screened. Those with positive NBS results performed at any health facility; 2) Unscreened. Those sick children clinically suspected of having IEiM who were hospitalized or ambulatory. In this category, the older unscreened affected siblings were also included.

Section 2.6: The word ‘registered’ is used multiple times within this section.  Perhaps on line four of the first paragraph, you could use the word ‘determined’.  Likewise in the second paragraph, ‘assessed’ might be substituted for ‘registered and analyzed’ in the first sentence.

Response: The suggestions were included.

Section 3: In the first line, you are not analyzing the individuals, but you are analyzing results from those individuals.  Simple clarification would be after the ’;‘ to include ‘data from’

Response: The suggestion was included in the corresponding section.

Section 3: Paragraph 7: Is ‘whether’ the word you intended or did you mean ‘if’ or ‘when’?  This reviewer would tend to use ‘when’ in that situation.

Response: Thank you for this suggestion. The change was made.

Discussion: First paragraph, line 3: Typographical error: ‘his’ should be ‘this’

Response: Thank you for this observation, the word was corrected.

Discussion: Paragraph 12: consider changing ‘but’ to ‘who’

Response: We agree, we changed the word.

Discussion Paragraph 13: Make the inserted new material ’these studies are also helpful..’ a new sentence

Response: Done.

Discussion: Paragraph 15: For clarity in English, on the next the last line change ‘once’ to ‘a time’

Response: Thank you, the change was made.

Conclusion: Paragraph 2 line one: Need an ‘a’ before national.

Response: Thank you, the change was made.
